# *Drosophila melanogaster* as a model for studies related to the toxicity of lavender, ginger and copaiba essential oils

Lucas Matos Martins Bernardes[1], Serena Mares Malta[1], Tamiris Sabrina Rodrigues[1], Luiz Fernando Covizzi[1], Rafael Borges Rosa[2,3], Allisson Benatti Justino[4], Renata Roland Teixeira[4], Foued Salmen Espíndola[4], Débora Oliveira dos Santos[5], Carlos Ueira Vieira[1], Murilo Vieira da Silva[2,3]*

1 Genetics Laboratory, Federal University of Uberlândia, Uberlândia, MG, Brazil, 2 Biotechnology in Experimental Models Laboratory, Federal University of Uberlândia, Uberlândia, MG, Brazil, 3 Rodents Animal Facilities Complex, Federal University of Uberlândia, Uberlândia, MG, Brazil, 4 Biochemistry and Molecular Biology Laboratory, Federal University of Uberlândia, Uberlândia, MG, Brazil, 5 Department of Oral and Maxillofacial Pathology, Federal University of Uberlândia, Uberlândia, MG, Brazil

* murilo.vieira@ufu.br

**Data Availability Statement:** All relevant data are within the paper and its Supporting Information files.

## Abstract

This study addresses the current trend of essential oils in alternative medicine using the non-chordate model *Drosophila melanogaster*. Following the three R's principles, it proposes non-chordate models to fill knowledge gaps on essential oil toxicity. Copaiba, lavender, and ginger essential oils are evaluated for effects on *D. melanogaster* lifespan, climbing ability, and brain structure, while their anti-inflammatory properties are also analyzed. Results show dose-related differences: higher concentrations (0.25% v/v) cause brain deterioration and impaired climbing, while lower concentrations (0.0625% v/v for copaiba and ginger; 0.125% for lavender) have no effect on climbing or brain structure. Lavender oil significantly extends lifespan and maintains anti-inflammatory activity when ingested, underscoring its therapeutic potential. These findings highlight the importance of *D. melanogaster* as a model for studying essential oil properties, potentially replacing chordate models. In addition, this research advances alternative remedies for currently incurable diseases, with lavender oil emerging as a promising candidate for drug discovery.

## Introduction

In 2021, the essential oils (EOs) market size was valued at $20.3 billion, with an estimated growth of 7.4% by 2028, due to their increasing use in industries such as food and beverages, cosmetics, and aromatherapy. In addition, because they are considered more "natural" than most commonly used medicines, their medical interest is increasing, and essential oils are slowly replacing drugs and medicines due to their reduced risk of side effects. Essential oils are basically substances extracted by distillation from different parts of plants, such as leaves, fruits, roots or flowers, and are composed of a mixture of hydrocarbons, ethers, esters, alcohols, aldehydes, phenols and terpenes [1,2].

**Funding:** The project was funded by the FAPEMIG agency (Project RED-00103-22). The funding was provided to support Lucas Matos Martins Bernardes as a scholarship holder. The funders did not participate in study design, data collection and analysis, decision to publish, or preparation of the manuscript.

**Competing interests:** The authors have declared that no competing interests exist.

Essential oils have a variety of different properties, including anti-inflammatory, antioxidant, antibiotic, and antiviral activities. The EOs extracted from lavender (*Lavandula angustifolia*), copaiba (*Copaifera reticulata, officinalis, coriacea, and langsdorffii*) and ginger (*Zingiber officinale*) all exhibited these activities both *in vitro* and *in vivo* [3–9], making them interesting targets for drug discovery that could be used to treat diseases that currently have no cure or treatment, such as neurodegenerative diseases and cancer. However, despite their widespread use and potential benefits, there is still a significant lack of information regarding the safety and efficacy of essential oils. Reports of adverse effects such as neurotoxicity, hepatotoxicity and reproductive abnormalities highlight the need for further research in this area [10–14].

To generate this knowledge, it is necessary to use animals as model organisms. However, this has been the subject of intense scientific debate due to ethical and legislative differences between countries. Data collected by the UK government in 2021 showed that more than 3 million procedures were performed on animals, of which 57% were experimental procedures, using animals in studies such as drug safety testing and the development of treatments [15].

Since the use of animals is essential, it is not always possible to avoid animal testing in order to obtain the same results. Considering the replacement principle of the "Three R's" guidelines established in 1959, the use of non-chordate animals (flies, worms) instead of chordates (mice, dogs, primates) is one of the possibilities to conduct research in a more ethical way, considering that non-chordates are less sentient animals [16]. The *Drosophila melanogaster* is a model organism that has already been used to test the toxicity of different compounds, such as metals and plant extracts [17,18]. Some of the advantages of using this animal are its fast life cycle, its cheap maintenance in laboratories and a genome with significant human homology [18–20]. Another advantage of using this model organism is the UAS-Gal4 system. First introduced in 1993, this system uses the yeast transcriptional activator Gal4, randomly inserted into the *D. melanogaster* genome, along with genes containing Gal4 binding sites, in this case UAS as an enhancer of the target gene. These genes are activated in cells expressing Gal4, allowing analysis of the phenotype produced by their expression. Expression of the yeast transcriptional activator could also be used alongside a driver gene to target the activation of target genes to specific organs or tissues [21].

In addition, the fruit fly model allows the evaluation of its behavior through assays already well established in the literature, such as the larval crawling assay, the courtship and mating assay, and the climbing assay, also known as the RING (Rapid Iterative Negative Geotaxis) test [22]. Because this test relies on the motor reflex of the flies, it can be used to detect, for example, neurodegeneration that might impair their motor abilities [23].

There have been some experiments using essential oils and fruit flies as a model organism, specifically the species *D. melanogaster* and *Drosophila suzukii* [24–26]. However, while these studies have analyzed the larvicidal and insecticidal effects of essential oils, our purpose on this study is to evaluate their toxicity and therapeutic potential in this model, aiming to propose this animal model as a replacement for chordate ones.

Therefore, this study aims to establish *D. melanogaster* as a viable model organism for studying the effects of essential oils in vivo. Specifically, we evaluate the toxicity and potential therapeutic effects of copaiba, lavender, and ginger essential oils on *D. melanogaster* lifespan, behavior, and brain morphology, as well as their anti-inflammatory activity, contributing to the scientific development and respecting the principles of the use of animals in research, especially the reduction of the number of chordate animals, as well as the replacement in many cases by a non-chordate in animal experiments.

## Materials and methods

### *Drosophila* stock and culture

*D. melanogaster* flies from the *Canton S* and $w^{1118}$ (Stock #3605) strains were obtained from the Bloomington Stock Center, Indianapolis, IN. The group also used the GMR-GAL4 > UAS-Eiger stock kindly provided to our laboratory by Dr. Masayuki Miura of the University of Tokyo (Tokyo, Japan).

Animals were reared on a standard cornmeal medium, and the tests were performed on a mashed potato medium containing 75% powdered mashed potato (Yoki®), 15% yeast extract, 9.3% glucose and 0.7% nipagin. The flies were maintained in a BOD incubator at a controlled temperature of 25˚C with a 12h/12h light/dark cycle. Flies were anesthetized with cold or ethyl ether for sex determination and mating.

Groups of 30 male adult $w^{1118}$ flies (0 to 3 days post-emergence) were used for all of the following tests, except for the toxicity evaluation where groups of 15 male and 15 female adult $w^{1118}$ flies (0 to 3 days post-emergence) were used. Each test lasted for 15 days, since the group aimed at a chronic treatment with the essential oils. All experiments were performed on three independent triplicates.

### Essential oils and solutions preparation

Essential oils of copaiba (*Copaifera sp.*), ginger (*Zingiber officinale*), and lavender (*Lavandula angustifolia*) (döTerra, Utah, USA) were used, diluted with water to different concentrations for each assay and treatment (0.0625% v/v, 0.125% v/v, 0.25% v/v, and 0.5% v/v), with the solution being thoroughly mixed by vortexing for 10 seconds to disperse the oil in the water each time it was pipetted. Both EOs and their dilutions were stored at room temperature and protected from light to prevent degradation. Chromatography charts of the EOs provided by döTerra can be found in the Supporting Information (S1–S3 Datasets).

### Toxicity evaluation of the essential oils

The first step, prior to treating the flies, was to determine the toxicity of each of the oils at different concentrations. Each essential oil was tested at concentrations of 0.0625% v/v, 0.125% v/v, 0.25% v/v, and 0.5% v/v. Groups of 30 adult $w^{1118}$ flies (0 to 3 days post-emergence) were separated into vials for each treatment in triplicate, with a 1:1 ratio of males to females.

Each vial contained mashed potato medium prepared with 5 mL of water (control group) or 5 mL of the EOs solutions. Flies were fed on the food prepared with these solutions for 15 days and transferred to vials containing fresh food every 2 to 3 days. The number of dead flies was counted each time they were transferred to a new vial.

### Lifespan analysis

To understand how consumption of the EOs would alter the lifespan of the flies, a lifespan assay was performed. Male flies of the $w^{1118}$ stock (0-to-1-day post-emergence) were separated into groups of 30 and fed on mashed potato medium prepared with the solutions at 0.0625% v/v for the copaiba and ginger EOs and 0.125% v/v for the lavender EO.

Similar to the toxicity evaluation, the flies were maintained on vials containing mashed potato medium prepared with 5 mL of water (control group) and 5 mL of the EOs solutions. Flies were transferred to vials containing fresh food every 2 to 3 days, and the number of dead flies was counted at each transfer until all flies had died.

## Climbing assay

The Rapid Iterative Negative Geotaxis (RING) test was used to determine whether the essential oils diet would cause any changes in the locomotor ability and behavior of the flies, performed as described by Gargano et. al in 2005 [27]. The only modification is that the digital image used to evaluate the position of the flies on the vials was taken 10 seconds after the apparatus was tapped on the surface. This is due to the strain of flies used, $w^{1118}$, which has a natural retinal degeneration that makes their climbing slower than other strains [28].

To understand how the animal model used would respond to an essential oil diet, the group performed the RING test using two sets of EO concentrations: lower doses that showed no toxicity in the toxicity test and higher doses that caused a greater number of deaths in the toxicity test. For the first, the concentrations used were 0.0625% v/v for the copaiba and ginger EOs and 0.125% v/v for the lavender EO. For the higher dose test, the concentrations used were 0.25% v/v for each essential oil.

The flies were fed on mashed potato medium prepared with the EO solutions at the above concentrations for 15 days, with the RING test performed every 5 days.

## Histological analysis

Knowing that the essential oils have small molecules that can cross the blood-brain barrier, the group decided to analyze whether ingesting the EOs could cause any damage to the flies' brains. Again, we decided to test both higher and lower doses of the essential oils to understand how the *D. melanogaster* model would respond to different concentrations of EOs.

Males from the $w^{1118}$ strain were collected from 0 to 1-day post-emergence and divided into groups of 10 flies in triplicates and fed on mashed potato medium prepared with the solutions of EOs. The animals were kept on this diet for 15 days, euthanized with liquid nitrogen and decapitated for preparation of brain tissue slides. The tissues were fixed in paraffin and stained with hematoxylin and eosin for better visualization, following the protocol used by Malta et. al 2022 [29].

The concentrations used were 0.25% v/v for each essential oil for higher doses and 0.0625% v/v for the copaiba and ginger essential oils and 0.125% v/v for the lavender essential oil for the lower doses.

## Analysis of the anti-inflammatory properties of the essential oils

Each of the essential oils used in this project has an anti-inflammatory activity already seen in both *in vitro* and *in vivo* tests [3–9]. However, the most commonly used animal models are mice and rats.

To analyze whether the *D. melanogaster* model would also respond to treatment with essential oils by ingestion, we used GMR-GAL4 > UAS Eiger flies as a model of inflammation in which eye degeneration is caused by overexpression of Eiger, a *Drosophila* ortholog of tumor necrosis factor alpha, in the organ. Assays in these animals are performed on larvae rather than adults because inflammation begins as soon as the animal hatches from the egg.

Flies of the GMR-GAL4 > UAS-Eiger strain were mated for 24 hours in oviposition medium. After this time, the parental flies were removed from the flask and the eggs were collected. These eggs were then placed in vials containing mashed potato medium prepared with solutions of the EOs at 0.025% v/v each. The animals were maintained on this medium for 10 days until the adult flies emerged from the pupae. The animals were then anesthetized with ice, and 60 flies from each group were collected for measurement of eye area using ImageJ software. The *Canton S* strain was used as a wild-type model for comparison of eye morphology.

### Statistical analysis

All statistical tests were performed using GraphPad Prism 8 software. The Mantel-Cox test was used to evaluate the statistical significance of the data for toxicity evaluation and lifespan analysis. Two-way ANOVA test was used for the climbing assay. One-way ANOVA test was used to analyze the anti-inflammatory properties of the essential oils. All values of $p < 0.05$ were considered significant.

## Results

### Essential oils demonstrate dose-dependent toxicity in *Drosophila* flies

Toxicity of essential oils was evaluated by exposing flies to food containing a mixture of water and oils at concentrations of 0.0625%, 0.125%, 0.25%, and 0.5% v/v for 15 days. At a concentration of 0.0625% v/v, lavender essential oil was the only oil that resulted in a higher mortality rate compared to the control group ($p < 0.05$) (Fig 1A). At a concentration of 0.125% v/v, flies exposed to copaiba and ginger essential oils had a mortality rate close to 50% ($p < 0.001$ and $p < 0.05$, respectively), while flies exposed to lavender EO had a mortality rate similar to the control (Fig 1B). At. 0.25% v/v, all three essential oils resulted in higher mortality compared to the control group (copaiba: $p < 0.05$, ginger: $p < 0.001$, and lavender: $p < 0.0001$) (Fig 1C). The result is similar at the 5% v/v concentration, where consumption of all essential oils resulted in more deaths compared to the control group ($p < 0.0001$) (Fig 1D).

### Safe concentrations of essential oils have varying effects on *Drosophila melanogaster* lifespan

To further investigate the effects of safe concentrations of essential oils on *Drosophila melanogaster*, we performed a lifespan analysis after prolonged exposure. We used concentrations of

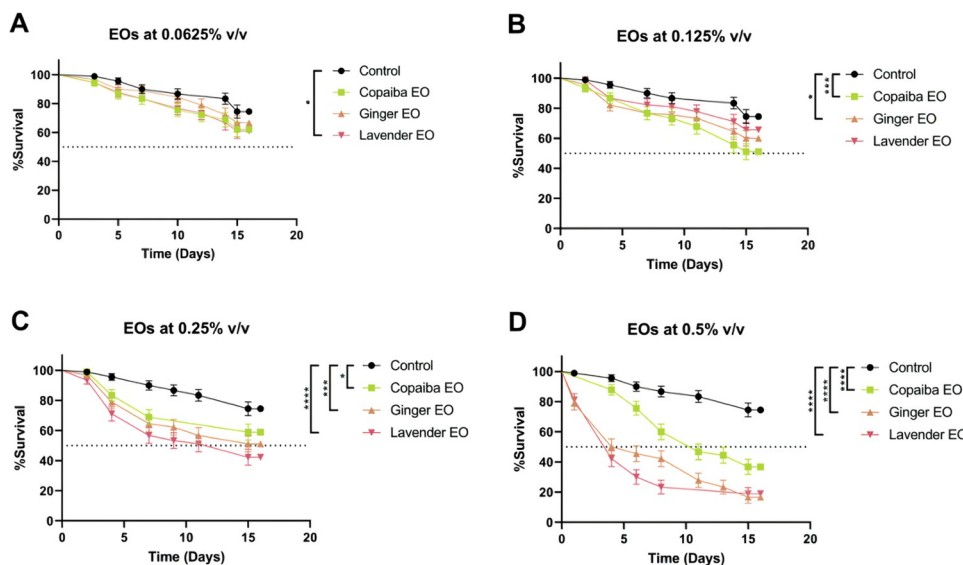

**Fig 1. Toxicity assessment of essential oils.** Lavender essential oil at 0.0625% v/v resulted in a higher mortality rate within 15 days of ingestion compared to the control group (A). At 0.125% v/v, copaiba and ginger essential oils caused more deaths, while no difference was observed for lavender EO (B). Both 0.25% and 0.5% v/v concentrations of EOs resulted in higher mortality rates than the control group (C and D). Statistical significance was determined by Mantel-Cox test (n = 30, * $p < 0.05$, *** $p < 0.001$, **** $p < 0.0001$). Error bars on the graphics represent the standard error. This test was performed on three independent triplicates.

0.0625% v/v for copaiba and ginger essential oils and 0.125% v/v for lavender essential oil. Our results showed that copaiba EO did not alter the lifespan or mortality rate of the flies compared to the control group. However, ginger EO caused a higher mortality rate in the flies from the fifteenth day, suggesting that long-term exposure may be toxic to the animals (p<0.01). In contrast, flies that ingested lavender EO had an increase in lifespan of 12 days compared to the control group (p<0.001) (Fig 2).

## Essential oils at low concentrations do not affect neuromotor function or brain morphology in flies

We evaluated the effects of safer concentrations of essential oils on neuromotor function and brain morphology in *Drosophila melanogaster*. The results showed that 0.0625% v/v concentration of ginger and copaiba essential oils and 0.125% v/v concentration of lavender essential oil did not affect the climbing ability of the flies. While copaiba EO had no effect on climbing ability, lavender EO increased climbing ability on days 5 (p<0.001) and 10 (p<0.05) of treatment, and ginger EO increased climbing ability only on day 5 (p<0.01) (Fig 3A–3C).

Furthermore, exposure to these lower doses did not cause any visible brain damage, as there were no vacuoles present in the brain tissue morphology of the treated flies compared to the control group (Fig 3D–3G).

## Essential oils cause vacuole formation in *Drosophila* brain and neuromotor impairment at higher concentrations

During the fifth and tenth days after treatment, the climbing ability of the flies remained relatively unaffected, except for the flies exposed to copaiba essential oil on day five (p<0.05) and ginger essential oil on day ten (p<0.05) (Fig 4A and 4B). However, by day 15, flies exposed to all three essential oils showed a 20% decrease in climbing ability compared to the control group, indicating neuromotor impairment caused by the EOs (copaiba and ginger EOs: p<0.01, and lavender EO: p<0.001) (Fig 4C).

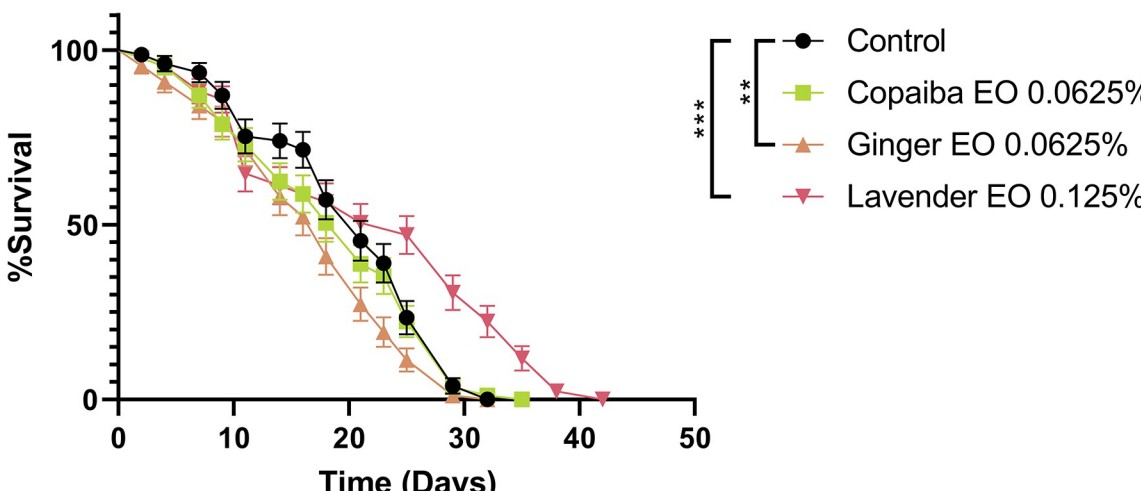

**Fig 2. Lifespan analysis of *D. melanogaster* ingesting essential oils.** Copaiba essential oil did not affect the lifespan of the flies, while ginger essential oil decreased it. However, lavender essential oil increased the lifespan by about 12 days. Statistical significance was determined by Mantel-Cox test (n = 30, ** p<0.01, *** p<0.001). Error bars on the graphics represent the standard error. This test was performed on three independent triplicates.

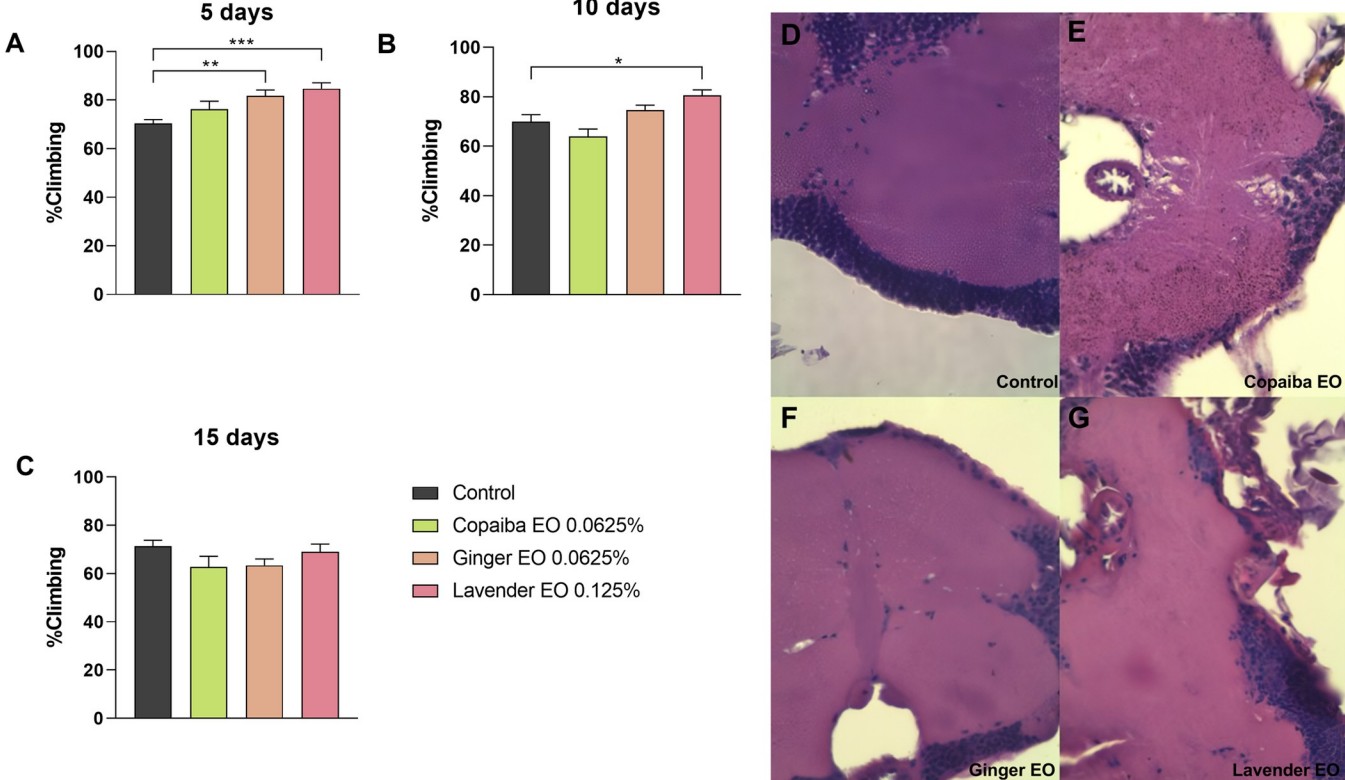

**Fig 3. Climbing assay and histological analysis of the brain at lower doses of essential oils intake.** Flies exposed to ginger and lavender essential oils showed improved climbing ability on day 5 (A), while only those treated with lavender essential oil continued to show improvement on day 10 (B). By day 15, there was no significant difference in climbing ability between the groups (C). Statistical significance was determined by two-way ANOVA test (n = 30, * $p < 0.05$, ** $p < 0.01$, *** $p < 0.001$), with graphics demonstrating mean ± standard error of the mean. Figures D to G illustrate the histologic section of the brain of flies from each group (D: control, E: copaiba EO, F: ginger EO and G: lavender EO). Histological analysis of the brain showed no morphological changes in any group. Images were taken at 100X magnification. This test was performed on three independent triplicates.

Histological analysis showed that a concentration of 0.25% v/v of the essential oils for 15 days caused neurological damage in the brains of *Drosophila melanogaster* flies. The presence of vacuoles in the brain tissue suggests that the essential oils may have caused damage to the nervous system, consistent with the neuromotor impairment observed in the climbing assay. The extent of brain damage varied among the three essential oils, with lavender EO producing more prominent vacuoles than the other two oils (Fig 4D–4G).

## Lavender essential oil promotes slight recovery of ocular inflammation caused by Eiger protein overexpression in *D. melanogaster*

A wild-type *D. melanogaster* eye phenotype, characterized by its large area and round shape (*Canton S* strain), is shown in Fig 5A. In contrast, Fig 5B shows the eye of an untreated GMR-GAL4 > UAS-Eiger fly, which shows degeneration due to overexpression of Eiger protein in the organ, resulting in loss of shape and area. Visual analysis of flies fed copaiba essential oil (Fig 5C) or ginger essential oil (Fig 5D) showed no significant evidence of reversal of ocular degeneration. However, when flies were fed lavender essential oil, a slight recovery of shape and increased eye area was observed (Fig 5E).

Quantitative analysis using ImageJ software revealed no statistical difference in eye area between the control group (untreated GMR-GAL4 > UAS-Eiger fly) and flies treated with

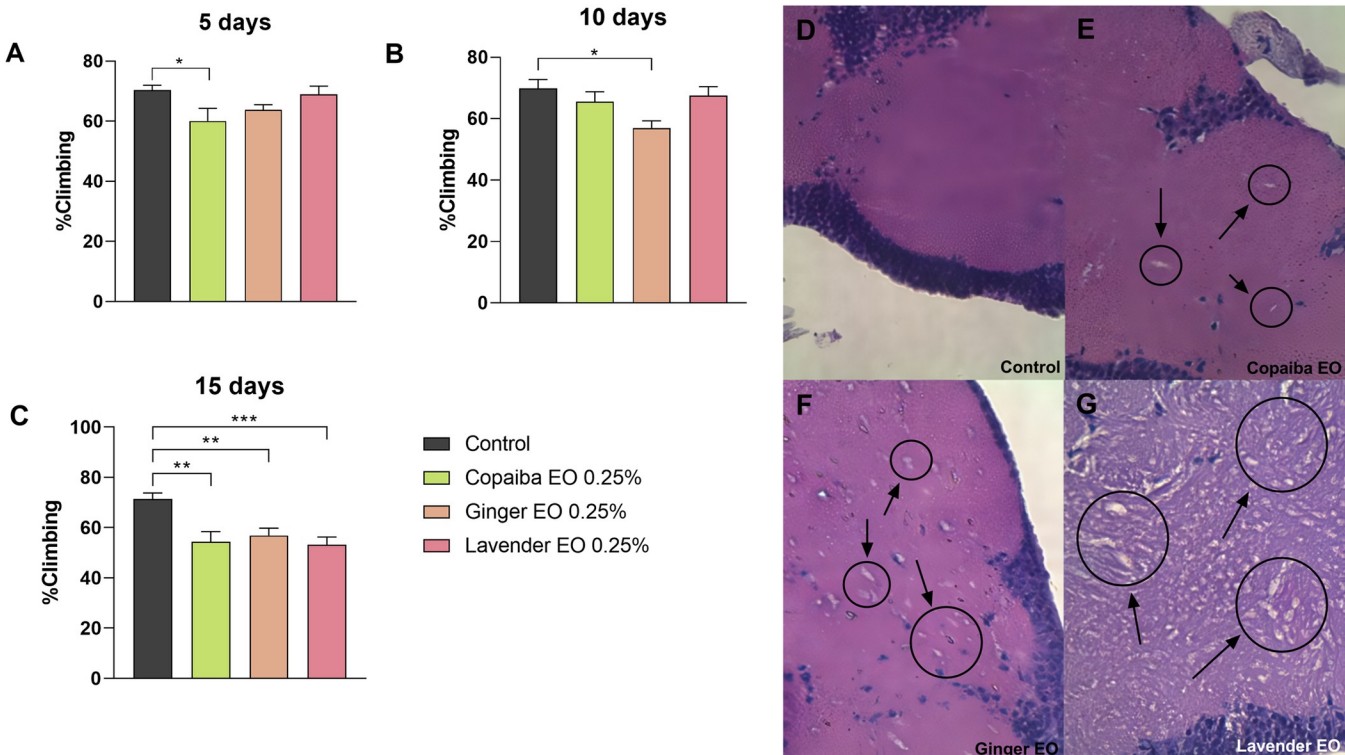

**Fig 4. Climbing assay and histological analysis of the brain at higher doses of essential oils.** On day 5 of treatment, only flies exposed to copaiba essential oil had reduced climbing ability, while on day 10, only flies exposed to ginger essential oil had this impairment. However, by day 15 of treatment, flies from all groups had reduced climbing ability compared to the control group. Statistical significance was determined by two-way ANOVA test (n = 30, * $p < 0.05$, ** $p < 0.01$, *** $p < 0.001$), **with graphics demonstrating mean ± standard error of the mean**. Figures D to G illustrate the histological section of the brain of flies from each group (D: control, E: copaiba EO, F: ginger EO, and G: lavender EO). The presence of vacuoles can be seen in each experimental group, particularly in those treated with lavender essential oil. Pictures were taken at 100X magnification. This test was performed on three independent triplicates.

copaiba essential oil or ginger essential oil. However, flies that ingested lavender essential oil had a significantly larger eye area compared to the control group ($p < 0.0001$) (Fig 5F).

## Discussion

The essential oils extracted from *Copaifera spp*, *Zingiber officinale* and *Lavender angustifolia* have shown promising results in previous studies regarding their anti-inflammatory, antioxidant and antimicrobial activities, as mentioned above. These results indicate their potential as alternative treatments for diseases and bacterial, fungal and viral control. However, their toxicity still needs to be evaluated in *in vivo* models to better understand their limitations as potential drugs. One way to fill this information gap is to test their use in model organisms, in this case *Drosophila melanogaster*.

In the experiments conducted, the *D. melanogaster* model showed changes in lifespan depending on the amount of essential oil they ingested. While at higher doses all EOs showed toxicity to the flies and reduced their lifespan, at lower doses they did not change it or, for lavender essential oil, increased it. These results indicate that this model can be a useful tool to assess the potential toxicity of essential oils. Furthermore, the observed increase in lifespan in flies that ingested lavender essential oil at lower doses highlights its potential as a natural product for further exploration in drug discovery.

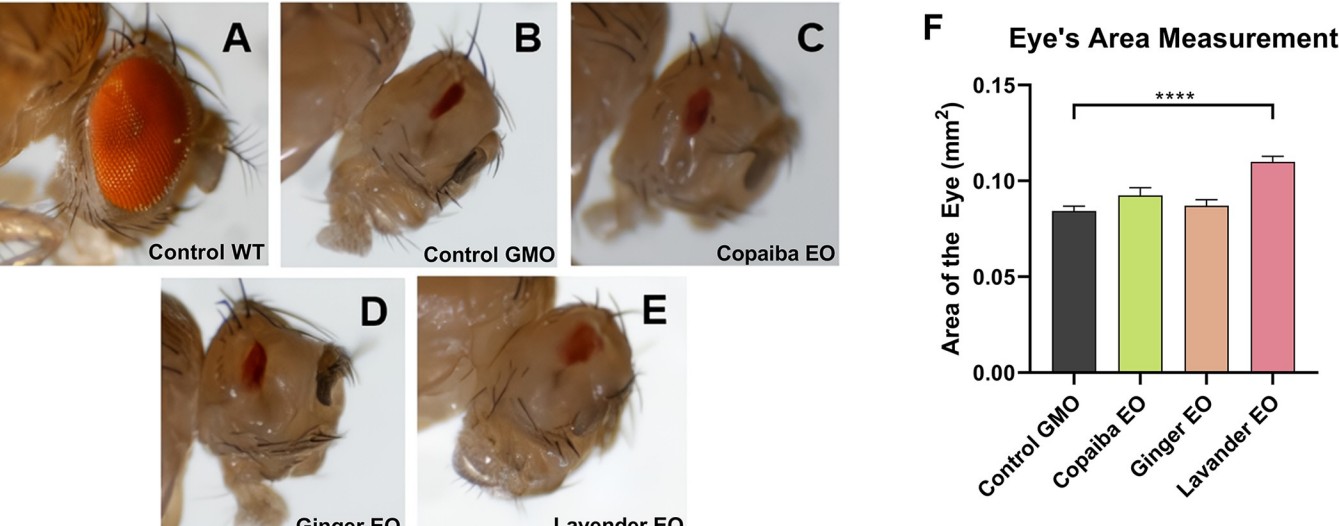

**Fig 5. Effect of essential oils on the eye area of flies.** Figures A through E show the eyes of flies from different groups. The Canton S group (A) had a normal eye, while the negative control group (B) with Eiger protein overexpression had a malformed eye, representing the GMR-GAL4 > UAS-Eiger strain. Experimental groups of GMR-GAL4 > UAS-Eiger flies treated with the essential oils at 0.25% v/v are shown in Figures C to E (C: copaiba EO, D: ginger EO, and E: lavender EO). Images were taken with a stereomicroscope at 3X magnification. Measurements taken with ImageJ show that only lavender essential oil was able to ameliorate the inflammation of the fly eye and slightly restore its area (F). Statistical significance was determined by one-way ANOVA test (n = 60, **** p<0.0001), with graphics demonstrating mean ± standard error of the mean. This test was performed on three independent triplicates.

Our investigation revealed that the ingestion of essential oils had a significant effect on the behavior of fruit flies, with climbing ability being particularly affected in a dose-dependent manner. Higher doses of essential oils resulted in impaired climbing ability, likely due to neurotoxicity, as evidenced by vacuoles observed in brain histology of affected flies. These results are consistent with previous studies showing that essential oils can have neurotoxic effects [30–32]. Our results demonstrate that fruit flies can be used to study the effects of essential oils on brain morphology and suggest that neurotoxicity is one of the possible adverse effects of overdosing on copaiba, ginger, and lavender essential oils.

At lower doses of essential oil intake, copaiba and ginger essential oils did not significantly affect the climbing ability of the flies during the 15-day treatment period. However, flies treated with lavender essential oil showed a slight improvement in their climbing ability, consistent with the histological analysis of their brains, where no major changes in morphology were observed. The natural eye degeneration of the $w^{1118}$ fly strain used in this study leads to slower climbing [28], making the improvement in climbing ability after lavender essential oil treatment particularly noteworthy. These results suggest that lavender essential oil may have a protective effect on retinal degeneration in *D. melanogaster*. However, further studies are needed to elucidate the underlying mechanisms and to confirm these findings.

Specifically for copaiba, studies analyzing the toxicity of copaiba oleoresin, which shares some of its constituents with the essential oil, found no acute or short-term toxicity in Wistar rats given the oleoresin by gavage at 2000 mg/kg (acute) and 100 mg/kg for 28 days (short-term) [33,34]. It is important to note that although they have the same constituents, essential oils are more concentrated and may have a different effect on animals compared to oleoresin. Unfortunately, studies analyzing the toxicity of essential oils are still scarce, especially in *D. melanogaster*, so more research is needed in this area.

While previous studies have demonstrated the anti-inflammatory activity of all three essential oils in various applications such as topical and inhalation [7,35–38], our investigation

revealed that only lavender essential oil had the potential to restore the area and shape of the eye after ingestion. This observation not only demonstrates the ability of flies to respond to essential oil treatment, but also suggests that lavender EO retains its anti-inflammatory properties even after ingestion, unlike ginger and copaiba EO. These findings provide opportunities for further research on lavender EO because several *D. melanogaster* strains can simulate human diseases for which there are no effective treatments or cures, including neurodegenerative diseases that involve inflammation as a pathological mechanism (e.g., Alzheimer's, Parkinson's, and Huntington's diseases). Notably, these *D. melanogaster* models are well established in the literature and serve as reliable sources of information on potential treatments for these diseases [39–41].

Nevertheless, this study has some limitations. The group analyzed changes in brain morphology which could be useful for future research on essential oils. However, further research is necessary to analyze the impact of essential oils on other organs, particularly the gastrointestinal tract. Moreover, the essential oils used have antimicrobial activity, which implies that they might have disturbed the flies' gut microbiota, as mentioned above. Dysbiosis of the microbiota is known to not only disrupt intestinal functions but also impact brain function via the gut-brain axis [42–47]. The group aims to investigate these questions in future research.

It is also important to mention that since the essential oils were administered to the flies through their food *ad libitum*, we could not ensure that the flies ingested the same amount of essential oil. A food intake analysis could be performed in the future to avoid this bias and to understand if the presence of the EOs on their food affects the amount of food they consume.

In conclusion, our results highlight the potential of *D. melanogaster* as a valuable model organism for evaluating the toxicity of essential oils. Our experiments revealed adverse effects of excessive doses of ginger, copaiba, and lavender essential oils, including neurotoxicity. Further investigations are warranted to explore their effects on other organs and their intestinal microbiota. However, our results also demonstrate the therapeutic potential of lavender essential oil, which not only prolonged the lifespan of *D. melanogaster*, but also induced an anti-inflammatory response without causing any damage to their brains. These observations suggest that lavender essential oil may be a promising candidate for the development of novel drugs.

## Conclusion

In conclusion, the experiments with *D. melanogaster* not only showed that it is a reliable model for studies with essential oils, but also that even though essential oils can have therapeutic potential, they also have adverse effects. While higher doses of essential oils caused toxicity and reduced the lifespan of the animals, lower doses did not cause any significant negative effect and even increased their lifespan, for the lavender essential oil. The main adverse effect observed was neurotoxicity resulting in impaired climbing ability.

Lavender essential oil, on the other hand, proved to be a promising natural product for drug discovery, with the potential to induce an anti-inflammatory response in the flies without causing damage to their brain. These findings highlight the usefulness of *D. melanogaster* as a model organism for studying the toxicity and therapeutic properties of essential oils, a novelty, given that most tests using this model and essential oils aim at identifying possible larvicidal and insecticidal effects, and indicate that further testing is needed to fully explore their effects on other organs and their potential application in drug development.

## Supporting information

**S1 Dataset. Chromatographic profile of copaiba essential oil.** Chromatographic analysis of the copaiba essential oil utilized in this study, including the lot number, chromatographic

chart, and a table of the constituents of the oil.
(PDF)

**S2 Dataset. Chromatographic profile of ginger essential oil.** Chromatographic analysis of the ginger essential oil utilized in this study, including the lot number, chromatographic chart, and a table of the constituents of the oil.
(PDF)

**S3 Dataset. Chromatographic profile of lavender essential oil.** Chromatographic analysis of the lavender essential oil utilized in this study, including the lot number, chromatographic chart, and a table of the constituents of the oil.
(PDF)

## Acknowledgments

We thank the members of the Genetics Laboratory and the Animal Facility Network of the Federal University of Uberlândia for their support during this project.

## Author Contributions

**Conceptualization:** Lucas Matos Martins Bernardes, Rafael Borges Rosa, Carlos Ueira Vieira, Murilo Vieira da Silva.

**Data curation:** Lucas Matos Martins Bernardes, Serena Mares Malta, Tamiris Sabrina Rodrigues, Luiz Fernando Covizzi, Allisson Benatti Justino, Renata Roland Teixeira, Foued Salmen Espíndola, Débora Oliveira dos Santos.

**Formal analysis:** Lucas Matos Martins Bernardes, Murilo Vieira da Silva.

**Funding acquisition:** Carlos Ueira Vieira, Murilo Vieira da Silva.

**Methodology:** Lucas Matos Martins Bernardes, Carlos Ueira Vieira, Murilo Vieira da Silva.

**Supervision:** Carlos Ueira Vieira, Murilo Vieira da Silva.

**Validation:** Lucas Matos Martins Bernardes, Carlos Ueira Vieira, Murilo Vieira da Silva.

**Writing – original draft:** Lucas Matos Martins Bernardes.

**Writing – review & editing:** Lucas Matos Martins Bernardes, Rafael Borges Rosa, Murilo Vieira da Silva.

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
