## [Decision Letter · Decision Letter 0]

12 Jul 2023

PONE-D-23-11609Drosophila melanogaster as a model for studies related to the toxicity of lavender, ginger and copaiba essential oilsPLOS ONE

Dear Dr. Rosa,

Thank you for submitting your manuscript to PLOS ONE. After careful consideration, we feel that it has merit but does not fully meet PLOS ONE’s publication criteria as it currently stands. Therefore, we invite you to submit a revised version of the manuscript that addresses the points raised during the review process.

Both reviewers found the manuscript interesting and merit in the data and the conclusions. However questions were raised and comments regarding methods, data presentation and manuscript presentation were raised.Please thoroughly address all comments and suggestions as they are aimed at improving the manuscript presentation and clarity

We look forward to receiving your revised manuscript.

Kind regards,

Efthimios M. C. Skoulakis, PhD

Academic Editor

PLOS ONE

“Yes. The project was funded by the Fapemig agency (Project RED-00103-22). The funding was given to pay scholarship holder Lucas Matos Martins Bernardes.”

Reviewers' comments:

Reviewer's Responses to Questions

**Comments to the Author**

1. Is the manuscript technically sound, and do the data support the conclusions?

Reviewer #1: Yes

Reviewer #2: Yes

2. Has the statistical analysis been performed appropriately and rigorously? 

Reviewer #1: Yes

Reviewer #2: Yes

3. Have the authors made all data underlying the findings in their manuscript fully available?

Reviewer #1: Yes

Reviewer #2: Yes

4. Is the manuscript presented in an intelligible fashion and written in standard English?

Reviewer #1: Yes

Reviewer #2: Yes

5. Review Comments to the Author

Reviewer #1: Essential oils have gained popularity for their potential therapeutic effects, but their safety and efficacy need to be scientifically evaluated. Overall, this study is significant because it provides scientific insights into the safety and efficacy of essential oils, offers an alternative model organism for research, highlights dosage considerations, and identifies lavender essential oil as a promising therapeutic agent. These findings have implications for both the scientific community and individuals interested in utilizing essential oils for health purposes, contributing to the advancement of knowledge and the development of alternative treatment options.

Here are my comments:

Introduction section

1) To strengthen their case, I suggest that the authors cite more papers that have utilized the drosophila model to study the effects of essential oils. Some suggested literature includes PMID: 36751763, PMID: 34865075, and PMID: 36234751.

2) I highly recommend that the authors add information about the gal4-uas system as one of the advantages in the sentence "Some of the advantages of using this animal are its fast life cycle, its cheap maintenance in laboratories, and a genome with significant human homology." This addition will provide a more comprehensive overview of the advantages of using the drosophila model.

Materials and methods section

1) Please add the following information, described in "Toxicity evaluation of the essential oils," to the "Drosophila stock and culture" section: "Groups of 30 adult w1118 flies (0 to 3 days post-emergence) were separated into vials for each treatment in triplicate, with a 1:1 ratio of males to females."

2) Please add the final concentrations after the dilutions of each essential oil, as described in the "Toxicity evaluation of the essential oils" section, to the following sentence: "Essential oils of copaiba (Copaifera sp.), ginger (Zingiber officinale), and lavender (Lavandula angustifolia) (döTerra, Utah, USA) were used, diluted with water to different concentrations for each assay and treatment."

3) I kindly request moving the sentence "Each vial contained mashed potato medium prepared with 5 mL of water (control group) or 5 mL of the EOs solutions. Flies were fed on the food prepared with these solutions for 15 days and transferred to vials containing fresh food every 2 to 3 days. The number of dead flies was counted each time they were transferred to a new vial. The Mantel-Cox survival test was used for statistical analysis." to the “ Lifespan analysis section."

4) It would be helpful if the authors could provide references for the statement: "Each of the essential oils used in this project has demonstrated anti-inflammatory activity in both in vitro and in vivo tests." to the Analysis of the anti-inflammatory properties of the essential oils section.

5) It is recommended to include a dedicated section in the Materials and Methods that describes the statistical analysis methods employed in this study. This would provide transparency and allow readers to understand how the data were analyzed and interpreted. Providing details such as the specific statistical tests used, any assumptions made, and the significance threshold would enhance the rigor and reproducibility of the study.

Results section

1) I recommend that the authors include the statistical significance values directly in the results section, in addition to the figure legends. This will provide readers with a clear and concise understanding of the statistical analysis and outcomes without having to refer back and forth between the figures and the accompanying text.

2) Did the authors investigate the larvicidal activity of the essential oils?

Discussion section

1) Authors must include a paragraph discussing the limitations of the study.

Figures:

1) Please increase the resolution of figures 1 (a, b, c and d), 3, 4 and 5.

2) Figure 3 - I highly recommend the authors to include the corresponding group names at the bottom of each histologic section of the brain (D: control, E: copaiba EO, F: ginger EO, and G: lavender EO). This will provide clarity and enable readers to easily associate each image with the respective treatment group. Additionally, it would be beneficial to indicate the image magnification (100X) to provide a sense of scale and facilitate a better understanding of the histological features.

3) Figure 4 - I highly recommend the authors to include the corresponding group names at the bottom of each histologic section of the brain (D: control, E: copaiba EO, F: ginger EO, and G: lavender EO). This will provide clarity and enable readers to easily associate each image with the respective treatment group. Additionally, it would be beneficial to indicate the image magnification (100X) to provide a sense of scale and facilitate a better understanding of the histological features. Additionally, I suggest that the authors consider including arrows in the histologic sections to clearly indicate the presence of vacuoles. This visual aid will help to emphasize and highlight the specific features being discussed in the image.

Reviewer #2: Comments regarding the manuscript are as follows:

1) Should be checked in italics and abbreviations in the manuscript. Abbreviations: At the first appearance in the abstract and the text, abbreviations should be preceded by words for which they stand.

2) Maybe authors should be added the keywords such as; genotoxicity, environment and health, risk assessment, etc.

3) I think that the concentrations of essential oils (EOs) of copaiba (Copaifera sp.), ginger (Zingiber officinale), and lavender (Lavandula angustifolia) (döTerra, Utah, USA) should be added to the Abstract.

4) Introduction: Authors should discuss the meaningfulness using Drosophila melanogaster compared to other in vivo model organisms. The flow of introduction need to be tuned.

5) The reason for using the in vivo model? What was the rationale aim for choosing D. melanogaster as a model organism? This information must be included in the Introduction section.

6) What is the significance of investigating EOs exposure routes?

7) The researchers need to recommend the dosage range which will be safe for human consumption.

8) I think that it should be mentioned in the Materials and Methods to reference studies for selected concentrations of EOs (v/v)? I think this should be explained and added in the manuscript in order to enable comparisons to other studies. How could they justify the concentrations selection for the study? Are these low or high concentrations? Working with high concentrations reduces the toxicological value of the results since they select the resistant individuals, or simply do not occur in real life. Is the dose limited by toxicity? The authors mention that in the Materials and methods.

9) The Materials and methods section should be referenced from the literature.

10) What's the likelihood for human exposure to used concentrations in EOs? Are they realistic exposure concentrations? Please explain the experimental rationale in discussion section. Please define better the relation between the tested dose and the potential environmental exposure. Moreover, the discussion should be improved with a more stressed comparative analysis of the own research results with literature works, already present.

11) The activation of toxicity mechanisms are not completely clear, and miss of any deep investigation.

12) What were the rationale for choosing the different in vivo assays? This must be included in the Introduction section.

13) How many independent experiments (excluding for replicates) were performed in this study? I think the authors should mention in the Materials and Methods section. Application period projected for groups were chosen according to what? The authors should fully clarify this point.

14) Section Discussion: This section also need to be improved with recent studies. i feel that some irrelevant discussion. Please discuss about the effect of EOs on D. melanogaster as an in vivo model.

15) What is the vehicle control name? Which were used group in the Statistical analysis?

16) What is positive control used for the assays? I think the authors should add values of positive control in the Figures.

17) The correlation between selected compound (or the viability) and the in vivo experiments?

18) In addition in the discussion section comparative evaluation with toxicological effects at cellular levels of EOs could add additional relevance to the work. Please define better the relation between the tested concentration and the potential environmental exposure. Moreover, the discussion should be improved with a more stressed comparative analysis of the own research results with literature works, already present.

19) The authors should include in the discussion some thoughts about how the in vivo findings can translate to human risk.

20) The authors should be added chemical constituents (%) (such as, CAS No, Quality %, Catalog number, etc.)

21) use abbreviations for SI units: d for day.

22) What do authors think about the effects of solubility of EOs on the cytotoxicity?

23) The authors should add some description about the disadvantages or difficulties of using animal in vivo and in vitro models related with EOs.

24) In Discussion: The authors should add new papers and update the literature review. More evidence on the link between toxicity and intestinal damage/life span caused by EOs could support the paper's idea (novelty) better.

25) The rationale for the selection of EOs must be stated at the end of the Discussion section.

26) Running title should be added.

27) In Figure legends, please indicate the following in each Figure legend: (a) the times of each experiments were repeated; (b) the number/replica of each group in each repeat; (c) are the data showed in the figure representative?

28) Please highlight the novelty aspect of the present research in the introduction, abstract and conclusion section.

29) Statistical analysis: The authors need to provide company of stats for the statistical software. There is not enough explanation in statistical analysis. What statistical tests were used in the manuscript? Arithmetic mean ± standard deviation (or arithmetic mean ± standard error) values should be shown in the Figures.

30) What were the rationale for choosing the different assays? This must be included in the Introduction section. Do the assays used by the authors are based on the OECD guidelines?

31) Reported images are really poor and need substantial improvement.

6. PLOS authors have the option to publish the peer review history of their article (what does this mean?). If published, this will include your full peer review and any attached files.

Reviewer #1: **Yes: **Ana Paula Mendes Silva

Reviewer #2: No

---

## [Author Response · Author response to Decision Letter 0]

23 Aug 2023

Dear reviewers,

Thank you for reading and reviewing our manuscript, facilitating the improvement of our research to a superior scientific standard. We noted the corrections that you presented, and we have implemented certain changes as a result. Please find attached the revised version of our manuscript, containing all the changes made.

In the following section, we will discuss each point made by the reviewers:

Reviewer #1:

Question 1: To strengthen their case, I suggest that the authors cite more papers that have utilized the drosophila model to study the effects of essential oils. Some suggested literature includes PMID: 36751763, PMID: 34865075, and PMID: 36234751.

Answer: We appreciate the suggestion and acted on it, as we believe it would enhance the quality of our Introduction section. The modifications are available on page 2.

Question 2: I highly recommend that the authors add information about the gal4-uas system as one of the advantages in the sentence "Some of the advantages of using this animal are its fast life cycle, its cheap maintenance in laboratories, and a genome with significant human homology." This addition will provide a more comprehensive overview of the advantages of using the drosophila model.

Answer: It is agreed that the rationale for the choice of Drosophila melanogaster as a model organism would be strengthened by including information on the Gal4-UAS system. Revisions to the Introduction section are located on page 2.

Question 3: Please add the following information, described in "Toxicity evaluation of the essential oils," to the "Drosophila stock and culture" section: "Groups of 30 adult w1118 flies (0 to 3 days post-emergence) were separated into vials for each treatment in triplicate, with a 1:1 ratio of males to females."

Answer: We appreciate the suggestion made by the reviewer and have implemented it. However, the group maintains that this information needs to appear in both the 'Toxicity evaluation of essential oils' and the 'Drosophila stock and culture' sections. Therefore, rather than moving the information to the second section, we have decided to add it to the first section mentioned. Please refer to page 4 for the changes made.

Question 4: Please add the final concentrations after the dilutions of each essential oil, as described in the "Toxicity evaluation of the essential oils" section, to the following sentence: "Essential oils of copaiba (Copaifera sp.), ginger (Zingiber officinale), and lavender (Lavandula angustifolia) (döTerra, Utah, USA) were used, diluted with water to different concentrations for each assay and treatment."

Answer: We appreciate the suggestion and agree that it would enhance the flow of the Materials and Methods sections. The group adopted the suggestion, and you can find the changes made on page 6.

Question 5: I kindly request moving the sentence "Each vial contained mashed potato medium prepared with 5 mL of water (control group) or 5 mL of the EOs solutions. Flies were fed on the food prepared with these solutions for 15 days and transferred to vials containing fresh food every 2 to 3 days. The number of dead flies was counted each time they were transferred to a new vial. The Mantel-Cox survival test was used for statistical analysis." to the “Lifespan analysis section.”

Answer: The group agrees that this information belongs in the “Lifespan Analysis” section, but also finds it relevant to the “Toxicity Evaluation of Essential Oils” section. Rather than relocating the information, the decision is to include it in the “Lifespan Analysis” section since the same methodology is used for both tests, while also ensuring that it can be found in the “Toxicity Evaluation of Essential Oils” section. You can find the changes in the mentioned section.

Question 6: It would be helpful if the authors could provide references for the statement: "Each of the essential oils used in this project has demonstrated anti-inflammatory activity in both in vitro and in vivo tests." to the Analysis of the anti-inflammatory properties of the essential oils section.

Answer: The group expresses gratitude for the suggestion and accepts it. While the references were already present in the Introduction section, we concur that their placement was inappropriate. We have also added the references to the Methodology section. The modifications can be located on page 8. Should the reviewer suggest further changes, we are open to accepting new comments.

Question 7: It is recommended to include a dedicated section in the Materials and Methods that describes the statistical analysis methods employed in this study. This would provide transparency and allow readers to understand how the data were analyzed and interpreted. Providing details such as the specific statistical tests used, any assumptions made, and the significance threshold would enhance the rigor and reproducibility of the study.

Answer: We appreciate the suggestion and agree that including a Statistics section is necessary for a clear analysis of our results and for reproducibility purposes. We added the section, and it is available on page 9.

Question 8: Did the authors investigate the larvicidal activity of the essential oils?

Answer: The main focus of our article was to test the therapeutic effects of essential oils. As such, we did not examine their larvicidal activity. We recognize that investigating the toxicity of essential oils to flies could be a crucial analysis. Given this, we appreciate your inquiry and will incorporate an assessment of the larvicidal activity of essential oil into our next project. This will help us enhance our future research on this essential topic.

Question 9: Authors must include a paragraph discussing the limitations of the study.

Answer: The group agrees that it is necessary to add a paragraph discussing the limitations of the study. We appreciate and accept the suggestion because it will strengthen our discussion. The new paragraphs are located on page 19.

Question 10: Please increase the resolution of figures 1 (a, b, c and d), 3, 4 and 5.

Answer: The group appreciates the suggestion and has made the figures clearer for better analysis of the results obtained. Modifications can be observed in the Results section. If additional adjustments are necessary, we would be pleased to make them.

Question 11: Figure 3 - I highly recommend the authors to include the corresponding group names at the bottom of each histologic section of the brain (D: control, E: copaiba EO, F: ginger EO, and G: lavender EO). This will provide clarity and enable readers to easily associate each image with the respective treatment group. Additionally, it would be beneficial to indicate the image magnification (100X) to provide a sense of scale and facilitate a better understanding of the histological features.

Answer: The figures and/or their subtitles now include both adjustments. Though we recognize the significance of magnification information, we opted to avoid placing it directly on the figure. Instead, we included the details on the subtitle to prevent the image from being too overloaded with information, potentially complicating the analysis of results. If the reviewer believes that the magnification information should be included in the figure, we can make the necessary alteration. Modifications are present in Figure 3.

Question 12: Figure 4 - I highly recommend the authors to include the corresponding group names at the bottom of each histologic section of the brain (D: control, E: copaiba EO, F: ginger EO, and G: lavender EO). This will provide clarity and enable readers to easily associate each image with the respective treatment group. Additionally, it would be beneficial to indicate the image magnification (100X) to provide a sense of scale and facilitate a better understanding of the histological features. Additionally, I suggest that the authors consider including arrows in the histologic sections to clearly indicate the presence of vacuoles. This visual aid will help to emphasize and highlight the specific features being discussed in the image.

Answer: The answer to Question 11 remains the same. Modifications have been made and can be located on Figure 4. The group will make any required updates if necessary.

Reviewer #2:

Question 1: Should be checked in italics and abbreviations in the manuscript. Abbreviations: At the first appearance in the abstract and the text, abbreviations should be preceded by words for which they stand.

Answer: Both italics and abbreviations were double-checked by the group, and we believe that all necessary adjustments have been made. We are happy to make any new correction that might be necessary to improve our article.

Question 2: Maybe authors should add the keywords such as; genotoxicity, environment and health, risk assessment, etc.

Answer: We appreciate the suggestion; however, we decided to keep the chosen keywords. While we recognize the potential use of “environment and health” and “risk assessment,” we believe that these concepts do not align with the article's main theme. Our work did not involve any genetics analysis; therefore, the term “genotoxicity” is irrelevant.

Question 3: I think that the concentrations of essential oils (EOs) of copaiba (Copaifera sp.), ginger (Zingiber officinale), and lavender (Lavandula angustifolia) (döTerra, Utah, USA) should be added to the Abstract.

Answer: The group acknowledges the suggestion, and we believe that it would enhance the quality of our abstract. Therefore, we accepted it and incorporated the suggested change. You can locate it on page two.

Question 4: Introduction: Authors should discuss the meaningfulness using Drosophila melanogaster compared to other in vivo model organisms. The flow of introduction needs to be tuned.

Answer: While the suggestion of discussing the utility of Drosophila melanogaster compared to other in vivo models is appreciated, we assert that our Introduction section clearly presents the reasons for selecting this model organism instead of the more commonly used chordates, as is explained on pages 3 and 4. This information is available on pages 3 and 4. We would be happy to add further details to enrich our article if the reviewer deems it necessary.

Question 5: The reason for using the in vivo model? What was the rationale aim for choosing D. melanogaster as a model organism? This information must be included in the Introduction section.

Answer: The justification for choosing D. melanogaster was already provided in the Introduction section on pages 3 and 4, as we have previously mentioned in response to Question 4. We selected this model to adhere to the Three R’s guidelines, as it allows us to substitute the usual chordate models with a non-chordate model, which helped us refine the experiments by studying the effects of these essential oils on this particular model. By doing so, we aim to reduce the number of chordate animals required for further research. In case the reviewer requests additional information, our group will gladly provide it.

Question 6: What is the significance of investigating EOs exposure routes?

Answer: Essential oils have three exposure routes: topical application, inhalation, and ingestion. The absorption and processing of essential oils depends on the route used. Therefore, it is important to understand the responses that each route produces in the model organism. The model organism utilized in this study is Drosophila melanogaster, which allows for the analysis of ingestion and inhalation through their food. For topical applications, however, chordate model organisms, such as mice or rats, would be more suitable for conducting necessary tests.

Question 7: The researchers need to recommend the dosage range which will be safe for human consumption.

Answer: While we appreciate the suggestion, the goal of our group for this project was to standardize the model and recommend the use of D. melanogaster as a model organism for essential oil screening tests. Although we obtained valuable data for this model, it is still essential to use chordate animal models for more detailed research that enables us to translate the findings to humans. Our primary objective was to recommend the use of fruit flies to decrease the number of chordate animals required. Therefore, recommending a more precise dosage range for humans will be possible after researching chordate models.

Question 8: I think that it should be mentioned in the Materials and Methods to reference studies for selected concentrations of EOs (v/v)? I think this should be explained and added in the manuscript in order to enable comparisons to other studies. How could they justify the concentration selection for the study? Are these low or high concentrations? Working with high concentrations reduces the toxicological value of the results since they select the resistant individuals, or simply do not occur in real life. Is the dose limited by toxicity? The authors mention that in the Materials and methods.

Answer: Few articles published have utilized these specific essential oils on the D. melanogaster model. As a result, we did not have a clear starting point for the concentrations to be used, resulting in the lack of references in our Materials and Methods section. We tested different doses and determined the fruit fly's response to each of them. Based on their toxicity on the model, we selected the concentrations to use.

Question 9: The Materials and methods section should be referenced from the literature.

Answer: The group recognizes the importance of providing references to the Materials and Methods section. Therefore, we accept the suggestion and have added references to that section.

Question 10: What's the likelihood for human exposure to used concentrations in EOs? Are they realistic exposure concentrations? Please explain the experimental rationale in the discussion section. Please define better the relation between the tested dose and the potential environmental exposure. Moreover, the discussion should be improved with a more stressed comparative analysis of the own research results with literature works already present.

Answer: The requested information is significant and requires evaluation in the future. However, given that our research was not focused on extending our findings to humans, we did not deeply analyze and compare the concentrations used to their possible environmental exposures. Moreover, since essential oils are used in diverse ways, it is inaccurate to define an average concentration to which individuals are exposed when using them. Further research is necessary to elucidate this information.

Question 11: The activation of toxicity mechanisms are not completely clear, and miss of any deep investigation.

Answer: Although we believe understanding the toxicity mechanism is important, it is not the focus of our project. Our primary aim in this study was to recommend using Drosophila melanogaster as a model organism for preliminary screening tests on essential oils. We propose this approach to refine the results before testing on chordate animals, thereby reducing the number of animals required. We plan to analyze the toxicity mechanisms in our future research, but we believe they are outside the scope of our current project.

Question 12: What were the rationale for choosing the different in vivo assays? This must be included in the Introduction section.

Answer: It is important to explain why we selected our in vivo assays in both the Introduction and Methodology sections. The group appreciates and acknowledges the suggestion. The changes implemented are located on page 4. If the reviewer feels the necessity to include more information, the group will be happy to provide it.

Question 13: How many independent experiments (excluding for replicates) were performed in this study? I think the authors should mention in the Materials and Methods section. Application period projected for groups were chosen according to what? The authors should fully clarify this point.

Answer: We added information concerning the number of independent experiments to the Materials and Methods section. We welcome the suggestion since it enhances our project. Find the changes on page 4 and in the subtitles for each figure in the Results section. To evaluate the toxicity and therapeutic effects under long-term treatment, we applied the essential oils for 15 days. Our forthcoming objective is to investigate the therapeutic effects of essential oils more profoundly in treating neurodegenerative diseases, specifically Alzheimer's disease, using long-term treatment. We did not include this information in the article. If the reviewer believes it is necessary, we will be happy to add it.

Question 14: Section Discussion: This section also need to be improved with recent studies. i feel that some irrelevant discussion. Please discuss about the effect of EOs on D. melanogaster as an in vivo model.

Answer: Regrettably, there is a significant lack of research using Drosophila melanogaster as a model organism for analyzing the therapeutic effects of essential oils, which is why more research is needed to fill this gap and why there are not many comparisons of our work with previous ones. The present research articles that can be located evaluate the larvicidal and insecticidal effects of essential oils on either D. melanogaster and/or Drosophila suzukii and are referred to in the Introduction section (page 4). In the event that the reviewer deems it necessary, we can also include these articles in the Discussion section.

Question 15: What is the vehicle control name? Which were used group in the Statistical analysis?

Answer: No vehicle was used during our experiments. The essential oil was directly added into the water. To allow for proper mixing, the mixture was vortexed thoroughly every time it was pipetted, to ensure the oil became evenly dispersed in the water.

Question 16: What is positive control used for the assays? I think the authors should add values of positive control in the Figures.

Answer: In light of our previously published work on D. melanogaster, the group believes that a positive control is necessary only for testing the anti-inflammatory activity of essential oils. However, our laboratory could not yet standardize a positive control for the UAS-eiger Drosophila stock we used. We are still searching for a compound that can serve as a positive control. So far, we have tested Leflunomide (6 mg/mL to 0,0117 mg/mL) and Methotrexate (0,75 mg/mL to 0,0058 mg/mL). At the time of writing this letter, additional tests are being conducted, assessing various concentrations of aspirin, ibuprofen, as well as different concentrations of leflunomide. We are excited to obtain the final results so that we can include a positive control in our future research based on these tests.

Question 17: The correlation between selected compound (or the viability) and the in vivo experiments?

Answer: It is uncertain what the reviewer meant with this question; therefore, we are unsure how to respond. We would be pleased to provide a proper answer if they have the option to rewrite it.

Question 18: In addition in the discussion section comparative evaluation with toxicological effects at cellular levels of EOs could add additional relevance to the work. Please define better the relation between the tested concentration and the potential environmental exposure. Moreover, the discussion should be improved with a more stressed comparative analysis of the own research results with literature works, already present.

Answer: As previously stated, our intention with this paper is not to compare the concentrations used for environmental exposure at this time (refer to Question 10). We have included a paragraph in our Discussion section (page 18) that compares our work to previous studies; however, they employ different in vivo models (Wistar rats), as few studies combine Drosophila melanogaster and essential oils. In particular, there is a scarcity of published papers investigating ginger essential oil, so we could not bring in other papers to compare our work with.

Question 19: The authors should include in the discussion some thoughts about how the in vivo findings can translate to human risk.

Answer: The required information has not been added to our Discussion in light of what was reported in Questions 10 and 18. Since our goal was not to translate our results to humans, we believe that this information would be disjointed in the Discussion. If the reviewer really sees the need and believes it would improve our publication, we could add it to the Discussion.

Question 20: The authors should be added chemical constituents (%) (such as, CAS No, Quality %, Catalog number, etc.)

Answer: The chromatography charts previously submitted as supplemental material contain information on the constituents of each essential oil, its catalog number, and its quality assurance. If the author feels the need to add new information, we would be happy to do so.

Question 21: Use abbreviations for SI units: d for day.

Answer: In our prior publications, we had a consistent format for the figures to ensure consistency. We refrained from including abbreviations in our graphs to maintain consistency. We aim to maintain the same pattern in this work, so we prefer not to alter the figures. If the reviewer deems it essential, we are willing to make changes accordingly.

Question 22: What do authors think about the effects of solubility of EOs on the cytotoxicity?

Answer: Based on our current results, our group is unable to provide an accurate answer to this question. Therefore, more research is needed to assess different concentrations of EO application and vehicles for comparison purposes. Although we aim to clarify this in the future, at present, we cannot provide a definitive answer without resorting to speculation.

Question 23: The authors should add some description about the disadvantages or difficulties of using animal in vivo and in vitro models related with EOs.

Answer: This information has not been included in our work because it is deemed inappropriate. We suggest using Drosophila melanogaster as an in vivo model for comparison with other models. This choice was made because an in vivo organism can provide us with comprehensive information about the processing of essential oils in the body, their positive and negative effects, and the possibility of conducting behavioral tests to support the given observations. If deemed necessary by the reviewer, this information can be added to the Introduction.

Question 24: In Discussion: The authors should add new papers and update the literature review. More evidence on the link between toxicity and intestinal damage/life span caused by EOs could support the paper's idea (novelty) better.

Answer: The group included additional papers in the Discussion section to enhance its content. The modifications related to the comparison of our work with previous research can be located on page 19. However, as previously stated, the scarcity of literature in this area limited our ability to compare/discuss our work properly.

Question 25: The rationale for the selection of EOs must be stated at the end of the Discussion section.

Answer: A new paragraph has been added to the Discussion section to explain the reasoning behind our selection of these particular essential oils. We appreciate your suggestion. The made modifications have been recorded on page 17.

Question 26: Running title should be added.

Answer: We are unsure of what is meant by the term 'running title'. Our research shows that it may refer to a shorter version of the title that appears on the top-left corner of each page. However, PLoS One does not specify the inclusion of a running title on their submission guidelines. If the reviewer deems it necessary, we could consider adding one.

Question 27: In Figure legends, please indicate the following in each Figure legend: (a) the times of each experiments were repeated; (b) the number/replica of each group in each repeat; (c) are the data showed in the figure representative?

Answer: The group appreciates the suggestion since it will make our results clearer and easier to understand; therefore, we accept it. Modifications can be located in the Results section.

Question 28: Please highlight the novelty aspect of the present research in the Introduction, abstract and conclusion section.

Answer: Though our Introduction already featured a paragraph quickly touching on the novelty of our research, we elaborated on this topic to strengthen it, as seen on page 4. This topic is also addressed in the Conclusion section. However, as the Abstract has a word-count limit, we were unable to include the topic without sacrificing the essential information it must convey. Therefore, we opted not to make any changes. We are receptive to any new modifications that the reviewer may deem necessary.

Question 29: Statistical analysis: The authors need to provide company of stats for the statistical software. There is not enough explanation in statistical analysis. What statistical tests were used in the manuscript? Arithmetic mean ± standard deviation (or arithmetic mean ± standard error) values should be shown in the Figures.

Answer: To enhance our statistical explanation, we included a paragraph, titled 'Statistical Analysis,' in the Materials and Methods section, providing further details on how the statistical analysis was conducted. We also added requested information to the figures’ subtitles. These changes are present on page 9, and in the figures within the Results section.

Question 30: What were the rationale for choosing the different assays? This must be included in the Introduction section. Do the assays used by the authors are based on the OECD guidelines?

Answer: The group employed assays commonly used in research papers incorporating Drosophila melanogaster as a model organism. We have appended the references to our methodology to enable the reviewer to cross-check them if needed. The team attempted to acquire information on OECD guidelines for research on Drosophila melanogaster, but the only guidelines obtained were for experiments on humans and mice/rat models. However, we have elaborated on the reasons for using the chosen assays, inscribing the explanation within our Introduction section to enrich clarity (page 4).

Question 31: Reported images are really poor and need substantial improvement.

Answer: We appreciate the suggestion. The group realized upon further inspection that the image quality was not up to our standards. The resolution of the images was heightened to improve clarity. If additional changes are necessary, kindly notify us so we can improve our work. The Results section reflects the changes made.

Best regards,

Lucas Matos Martins Bernardes

Rafael Borges Rosa

Murilo Vieira da Silva

---

## [Decision Letter · Decision Letter 1]

25 Aug 2023

Drosophila melanogaster as a model for studies related to the toxicity of lavender, ginger and copaiba essential oils

PONE-D-23-11609R1

Dear Dr. Rosa,

We’re pleased to inform you that your manuscript has been judged scientifically suitable for publication and will be formally accepted for publication once it meets all outstanding technical requirements.

Kind regards,

Efthimios M. C. Skoulakis, PhD

Academic Editor

PLOS ONE

Additional Editor Comments (optional):

Reviewers' comments:

Reviewer's Responses to Questions

**Comments to the Author**

1. If the authors have adequately addressed your comments raised in a previous round of review and you feel that this manuscript is now acceptable for publication, you may indicate that here to bypass the “Comments to the Author” section, enter your conflict of interest statement in the “Confidential to Editor” section, and submit your "Accept" recommendation.

Reviewer #1: All comments have been addressed

Reviewer #2: All comments have been addressed

2. Is the manuscript technically sound, and do the data support the conclusions?

Reviewer #1: Yes

Reviewer #2: Yes

3. Has the statistical analysis been performed appropriately and rigorously? 

Reviewer #1: (No Response)

Reviewer #2: Yes

4. Have the authors made all data underlying the findings in their manuscript fully available?

Reviewer #1: Yes

Reviewer #2: Yes

5. Is the manuscript presented in an intelligible fashion and written in standard English?

Reviewer #1: Yes

Reviewer #2: Yes

6. Review Comments to the Author

Reviewer #1: The authors have diligently taken into account all of the comments, and I am confident that the manuscript is now in a commendable state for acceptance and publication.

Reviewer #2: I have checked the revised manuscript (Manuscript ID: PONE-D-23-11609R1, the paper entitled “Drosophila melanogaster as a model for studies related to the toxicity of lavender, ginger and copaiba essential oils”) and responses and I do not see anything critically wrong with this manuscript. The author has responded to the comments and made the revisions accordingly. Thus this manuscript can be accepted for publication.

7. PLOS authors have the option to publish the peer review history of their article (what does this mean?). If published, this will include your full peer review and any attached files.

Reviewer #1: **Yes: **Ana Paula Mendes-Silva

Reviewer #2: No

---

## [Editor Report · Acceptance letter]

19 Sep 2023

PONE-D-23-11609R1 

*Drosophila melanogaster* as a model for studies related to the toxicity of lavender, ginger and copaiba essential oils 

Dear Dr. Vieira da Silva:

I'm pleased to inform you that your manuscript has been deemed suitable for publication in PLOS ONE. Congratulations! Your manuscript is now with our production department. 

Kind regards, 

on behalf of

Dr. Efthimios M. C. Skoulakis 

Academic Editor

PLOS ONE